# Volatile Profiles of Five Variants of *Abeliophyllum distichum* Flowers Using Headspace Solid-Phase Microextraction Gas Chromatography–Mass Spectrometry (HS-SPME-GC-MS) Analysis

**DOI:** 10.3390/plants10020224

**Published:** 2021-01-24

**Authors:** Yeong-Geun Lee, Won-Sil Choi, Seung-Ok Yang, Jeon Hwang-Bo, Hyoun-Geun Kim, Minzhe Fang, Tae-Hoo Yi, Se Chan Kang, Youn-Hyung Lee, Nam-In Baek

**Affiliations:** 1Department of Oriental Medicine Biotechnology, Graduate School of Biotechnology, Kyung Hee University, Yongin 17104, Korea; lyg629@nate.com (Y.-G.L.); hbj3286@khu.ac.kr (J.H.-B.); zwang05@naver.com (H.-G.K.); mincheolbang1030@gmail.com (M.F.); drhoo@khu.ac.kr (T.-H.Y.); sckang@khu.ac.kr (S.C.K.); 2National Instrumentation Center for Environmental Management, Seoul National University, Seoul 08826, Korea; choialla@snu.ac.kr (W.-S.C.); soyang@snu.ac.kr (S.-O.Y.); 3Department of Horticultural Biotechnology, Kyung Hee University, Yongin 17104, Korea; younlee@khu.ac.kr

**Keywords:** *Abeliophyllum distichum*, HS-SPME-GC-MS, morphological characteristics, scents, volatile components analysis

## Abstract

*Abeliophyllum distichum* (Oleaceae), which is the only species in the monotypic genus and is grown only on the Korean peninsula, has a high scarcity value. Its five variants (white, pink, round, blue, and ivory) have different morphological characteristics in terms of the color of petals and sepals or shape of the fruits. Despite its high value, there has been no study on variant classification except in terms of their morphological characteristics. Thus, we performed a volatile component analysis of *A. distichum* flowers and multivariate data analyses to reveal the relationship between fragments emitted from five variants of *A. distichum* flowers with their morphological characteristics. As a result, 66 volatile components of this plant were identified by headspace solid-phase microextraction gas chromatography–mass spectrometry (HS-SPME-GC-MS), showing unique patterns for each set of morphological characteristics, especially the color of the petals. These results suggest that morphological characteristics of each variant are related to the volatile composition.

## 1. Introduction

Flower fragrances and pigments are characteristics of various insect-pollinated flowers; they serve as a major signal to lure pollinating insects to the reproductive organs. [1,2]. The fragrances of various flowers (e.g., lilac, rose, jasmine) have been used in perfumery due to their pleasant effects on the human sensory system, and some of them have been synthesized for production of artificial perfumes [3]. Thus, much research has focused on identifying and characterizing their odors and flavors.

After approval of the Nagoya Protocol, research using domestic native species became a key issue [4]. Among the various Korean plants, *Abeliophyllum distichum* (Oleaceae), which is the only species in a monotypic genus (*Abeliophyllum*) and is grown only in the Korean peninsula, has high scarcity value [5]. For this reason, the Korean government designated *A. distichum* as an endangered species until recently, and have even designated some of its habitats as natural monuments to preserve them [6]. Due to its high scarcity and ecological and geographic value, *A. distichum* has rarely been studied compared with other plants. However, in 2017, *A. distichum* was removed from the list of endangered species by the Ministry of the Environment [6] due to the development of mass breeding techniques [7,8]. Previous phytochemical studies of these plants have focused on just the leaves, and they reported that four phenylethanoid glycosides and two flavonoids are components of these plants [9,10,11]. These compounds have been reported to have anti-oxidation, anti-inflammation, anti-hypertension, anti-diabetes, and neuroprotection effects [10,11,12,13].

Moreover, this plant is known to have attractive and strong fragrances [14,15]. Even though *A. distichum* has a good fragrance, few studies have reported analyses of the associated volatiles. If such data were available, these volatiles could be manufactured as fragrance oil for various applications such as in aromatherapy, perfumes, and cosmetics. Thus, the aim of our study was to reveal the biological effects of *A. distichum* volatiles and standardize its content for use by the cosmeceutical industry. Accordingly, volatile analytes of *A. distichum* have high value from both the research and industrial perspectives.

Various quantitative and qualitative protocols for determining the volatile compositions in the flowers have been developed using their essential oils (such as Soxhlet extraction) [16,17]. However, these protocols have several disadvantages, such as loss of fragrance, production of artificial volatiles during solvent extraction, and consumption of a large number of organic solvents and time [16,17].

To compensate for these shortcomings, headspace analysis is used as a preferred analytical protocol for the volatile composition of natural materials [18]. However, there are some problems associated with headspace methods such as memory effects on traps and dynamic purging. For this reason, solid-phase micro extraction (SPME), which is a simple and rapid sample preparation technique, was developed by Arthur in 1990 [19]. This method has the powerful advantages of solvent-free extraction as well as concentration in a single step method [20]. Thus, it has been widely used for fragrance analysis [21]. Therefore, we performed a volatile analysis of *A. distichum* flowers using HS-SPME coupled with gas chromatography–mass spectrometry (GC-MS). To identify volatile components, we measured mass spectra for each peak and compared them with a mass spectral database (NIST library) and LRI (Linear retention indices) values [22,23].

Moreover, there have been some reports showing that flower volatiles are related to environmental factors, biological recognition, and morphological characteristics [24,25]. At present, five variants of *A. distichum* have been reported: white miseon (*A. distichum* Nakai), pink miseon (*A. distichum* for. *lilacinum* Nakai), ivory miseon (*A. distichum* for. *eburneum* T. B. Lee), blue miseon (*A. distichum* for. *viridicalycinum* T. B. Lee), and round miseon (*A. distichum* var. *rotundicarpum* T. B. Lee) [26,27,28]. These variants have different morphological characteristics in terms of the color of petals and sepals or fruit shapes, respectively.

In this study, HS-SPME-GC-MS analysis was performed to estimate the differences in fragrance composition of volatiles such as monoterpenes, alcohols, and aromatics on the basis of the morphological characteristics of the five variants of *A. distichum* flowers (white, pink, ivory, blue, and round miseon). In addition, data of volatiles emitted from five variants were further interpreted on the basis of multivariate data analyses to visually characterize the dissimilarities associated with morphological characteristics (especially the color of the petals). These analyses included principal component analysis (PCA) and partial least squares discriminant analysis (PLS-DA).

## 2. Results and Discussion

### 2.1. Profiling of Volatile Components from Five Variants of the Flowers in Five Variants of A. distichum Using HS-SPME-GC-MS Analysis

Identification of volatiles was conducted on the basis of the total ion chromatogram (TIC) from the measurement of five variants of *A. distichum* flowers using HS-SPME (Thermo Scientific Trace™ 1300, Thermo Fisher Scientific Inc., Sunnyvale, CA, USA) coupled with triple quadrupole mass spectrometer (QqQ-MS, TSQ 8000, Thermo Fisher Scientific Inc., USA) (Appendix A). The intensities of peaks, especially those eluted from 30–35 and 40–45 min, were significantly different depending on the sample. As shown in Appendix A and Table 1, each variant had different fragrance compositions as well as different morphological characteristics (such as color of petals and sepals and fruit shape). The TIC spectra for each variant were deconvoluted using Xcalibur 3.1 software from the Thermo Finnigan Corporation, San Jose, CA, USA. Each peak was identified by matching its spectrum with the NIST library (>85% similarity) and by analyzing the retention indices calculated against *n*-alkanes (C_7_–C_30_) (i.e., retention time and relative retention time). The resulting identifications of peaks were confirmed through an analysis of fragmentation patterns in the mass spectra. A total of 66 volatiles were identified and quantified in the headspace of 5 variants of *A. distichum* flowers, including 16 oxygenated monoterpenes, 1 oxygenated sesquiterpene, 5 monoterpene hydrocarbons, 12 aromatic alcohols, 5 aromatics, 13 alcohols, 10 aldehydes and carboxyls, and 4 miscellaneous components grouped as “others” (Table 1 and Appendix A). The structure types and the odor notes of each scent are illustrated in Appendix A.

### 2.2. Multivariate Data Analyses of Volatile Components from Five Variants of A. distichum Flowers

To understand how the classification of volatile components emitted from five variants of *A. distichum* flowers could be expressed and correlated to morphological characteristics, we performed multivariate data analyses (PCA and PLS-DA) for the scents identified in Section 2.1. Volatile component fingerprinting of five variants of *A. distichum* flowers (white miseon, pink miseon, ivory miseon, blue miseon, and round miseon) (Appendix A) was carried out with PCA, a method of unsupervised multivariate projection, to effectively describe the dissimilarities on the basis of the volatile data patterns [29]. Mathematical methods of par-scaling and mean-centering the datasets resulting from the above samples to unit variance were performed using the SIMCA-P 14.1 software. Before multivariate data analyses, each peak with an identified volatile (Appendix A) was normalized to the internal standard (1,2-dichlorobenzene-*d*_4_).

The PCA and PLS-DA results showed distinct separations between all five variants of *A. distichum* flowers, indicating that volatile components are related to the phenotype of each variant. The PCA result derived from the volatiles of each variant showed a score plot (Figure 1) together with a loading plot (Appendix A) consisting of PC1 (50.8%) and PC2 (27.8%). These described 78.6% of the total variance in the optimal separation of data. White miseon and ivory miseon and others were clearly separated by the principal PC1 and by the PC2, while white, pink, and blue miseon were clearly separated from the others. The loading plots (Appendix A) of PCA results were consistent with their respective score plots. The described dissimilarities of aroma characteristics relative to their correlative clusters highlighted variations in their fingerprints. On the basis of PC1 in the loading plot, white and ivory miseon showed higher contents of three oxygenated monoterpenes (l-linalool, linalool oxide, and epoxylinalool), two monoterpene hydrocarbons (*β*-myrcene and ocimene), and six alcohol derivatives (ethanol, 1-hexanol, 2-hexenal, 2-hexenol, (*Z*)-3-hexenol, and *n*-hexyl acetate) than others. On the other hand, one oxygenated monoterpene (hotrienol), three aromatic alcohols (benzeneethanol, methyl salicylate, and benzaldehyde), and one aldehyde (isovaleraldehyde) were higher in pink, blue, and round miseon compared with others. On the basis of PC2 in the loading plot, 10 oxygenated monoterpenes (epoxylinalool, hotrienol, lilac alcohol B–D, lilac aldehyde A–C, linalool oxide, and l-linalool), one monoterpene hydrocarbon (*β*-myrcene), two aromatic alcohols (benzeneethanol and methyl salicylate), and two alcohol derivatives (ethanol and nonaldehyde) were higher in white, pink, and blue miseon compared with others. In addition, two alcohol derivatives (2-hexenol and *n*-hexyl acetate) were highly concentrated in ivory miseon, round miseon, and blue miseon.

PLS-DA analysis was performed to further confirm the segregation of volatiles emitted from each variant [30]. Analogously, PLS-DA results (Figure 2 and Appendix A), in which the total variance was 78.6%, showed the same behavior as the PCA results. This alignment indicates that volatile fingerprints are deeply related to the morphological characteristics of *A. distichum* flowers.

On the basis of the variable influence on projection (VIP; Table 2), we found that 30 volatiles from *A. distichum* flowers were higher than 0.7, and 18 scents showed VIPs larger than 1.0, which is the most influential value for this model [31]. Table 2 indicates that the scents of plants, especially oxygenated monoterpenes, were related to separation in each variant. These results suggest that morphological characteristics of each variant were related to the content of volatiles.

## 3. Materials and Methods

### 3.1. Plant Materials

The flowers of *A. distichum* were provided by the Miseonnamu-maeul Agricultural Association Corporation (CEO: Jongtae Woo), Goesan-gun, Chungcheongbuk-do, Republic of Korea, in March 2017. Five variants, white miseon (*A. distichum* Nakai, KHU-NPCL-201704-01), pink miseon (*A. distichum* for. *lilacinum* Nakai, KHU-NPCL-201704-02), ivory miseon (*A. distichum* for. *eburneum* T. B. Lee, KHU-NPCL-201704-03), blue miseon (*A. distichum* for. *viridicalycinum* T. B. Lee, KHU-NPCL-201704-04), and round miseon (*A. distichum* var. *rotundicarpum* T. B. Lee, KHU-NPCL-201704-05), were identified by Prof. Dae-Keun Kim, College of Pharmacy, Woosuk University, Jeonju, South Korea. Voucher specimens (KHU-NPCL-201704-01-05) have been deposited at the Natural Products Chemistry Laboratory, Kyung Hee University.

### 3.2. Volatile Component Analysis from Five Variants of A. distichum Flowers

Approximately 30 g of raw floral tissue of 5 variants of *A. distichum* were collected between 10:00 a.m. to 10:30 a.m. on 25 March 2017 (12 ± 1 °C; 46 ± 2% relative humidity). The moistened samples were immediately transported to the laboratory. Fresh tissue of 5 variants of *A. distichum* flowers (1.0 g) were transferred into 20 mL glass vials sealed by metal screw-caps with pre-notched Teflon silicone septa. Approximately 10 flowers were used for each sample. Then, 5 μL of deuterated 1,2-dichlorobenzene-*d*_4_ (1000 ppm in pyridine) was added in each sample as an internal standard. Volatile components were extracted by HS-SPME using a 2 cm 50/30 μm DVB/CAR/PDMS StableFlex fiber (Supelco, Bellefonte, PA, USA). The samples were incubated and shaken for 20 min (using a 10 s on/off cycle) at 50 °C, with the fiber introduced halfway through this period (TriPlus RSH with SPME module; Thermo Scientific, USA). Other SPME parameters were set as follows: extraction time: 40 min, desorption time: 2 min, pre-conditioning time: 5 min, post-conditioning time: 20 min. Subsequently, the fiber was inserted into the injector port of the GC (Thermo Scientific Trace 1300, Thermo Fisher Scientific Inc., USA) for 1 min at 250 °C. A 1 μL of sample was injected in split mode (20:1, *v/v*). The volatile compounds were separated using a DB-WAX column (60 m × 0.25 mm internal diameter × 0.5 μm film thickness). Helium was used as the carrier gas at a constant flow rate of 3.0 mL/min. The oven temperature profile was started at 40 °C for 3 min, and then was programmed to reach 250 °C at a rate of 4 °C/min, staying at 250 °C for 5.5 min. The GC-MS transfer line temperature was set to 270 °C. A triple quadrupole mass spectrometer (TSQ 8000, Thermo Fisher Scientific Inc., USA) operated in scan mode at 70 eV with the electron ionization (EI) source kept at 250 °C. Five scans per second were recorded over the mass range *m/z* 35–550. The identification of volatile compounds was confirmed by comparing their spectra and retention times with standards from the library NIST 2.0. Quantitative analysis by the peak area normalization method was conducted to determine the relative amounts using internal standard. Analysis of each sample was performed three times. Linear retention indices (LRI) were calculated for each respective mass spectrum using the following equation: LRI = 100 × n + [100 × (*t*_x_ − *t*_n_)]/(*t*_n+1_ − *t*_n_), where x is the targeted compound x, n is the number of carbon atoms of the *n*-alkane eluted before x, n + 1 is the number of carbon atoms of the *n*-alkane eluted after x, *t*_x_ is the retention time of x, *t*_n_ is the retention time of n, and *t*_n+1_ is the retention time of n+1 [32].

### 3.3. Multivariate and Statistical Analysis

Raw chromatographic data acquired from triple quadrupole GC-MS analyses were processed by Xcalibur 3.1 software (Thermo Finnigan Corporation, San Jose, CA, USA), in which automatic peak detection and mass spectrum deconvolution (compound identification) were performed with references to library NIST 2.0. PCA and PLS-DA were then chosen to create a prediction model. SIMCA version 14.1 (Umetrics, Umeå, Sweden) and MetaboAnalyst 4.0 (http://www.metaboanalyst.ca) were initially employed to comprehend the relationship in terms of similarity or dissimilarity among groups of multivariate data. Statistical analysis was performed using a GraphPad Prism software version 7.00 (GraphPad Software, Inc., San Diego, CA, USA). Significance was estimated using repeated one-way analysis of variance (ANOVA) followed by Tukey’s test. Data were presented as mean ± standard error.

## 4. Conclusions

At present, five variants of this plant have been reported: white miseon (*A. distichum* Nakai), pink miseon (*A. distichum* for. *lilacinum* Nakai), ivory miseon (*A. distichum* for. *eburneum* T. B. Lee), blue miseon (*A. distichum* for. *viridicalycinum* T. B. Lee), and round miseon (*A. distichum* var. *rotundicarpum* T. B. Lee) [26,27,28]. The variants were classified only on the basis of morphological characteristics such as the color of the petals and sepals or the shapes of the fruit. There are many opinions on the taxonomic identities of variants of *A. distichum*, and some documents suggest that each variant was the same taxa [33]. Accordingly, a phytochemical investigation and chemical mapping of the variants is valuable.

In this study, we performed volatile component analysis of *A. distichum* flowers and multivariate data analyses to reveal the relation between fragments emitted from five variants of *A. distichum* flowers with their morphological characteristics. Sixty-six volatile components of this were plant identified and quantified by HS-SPME-GC-MS. Multivariate data analyses showed that almost all volatiles had unique patterns for each set of morphological characteristics, especially the color of the petals. These results suggest that the morphological characteristics of each variant were related to the content of the volatiles. Moreover, further studies are needed to understand the chemical origin of the morphological characteristics of each variant to reveal the dissimilarities of primary and secondary metabolites of these plants.

## Figures and Tables

**Figure 1 plants-10-00224-f001:**
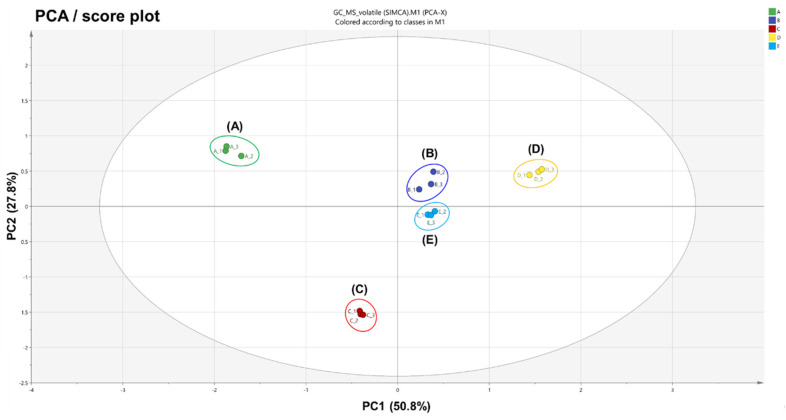
Principal component analysis (PCA) score plot obtained from HS-SPME-GC-MS results on five variants of *Abeliophyllum distichum* flowers: (**A**) white miseon, (**B**) pink miseon, (**C**) ivory miseon, (**D**) blue miseon, (**E**) round miseon.

**Figure 2 plants-10-00224-f002:**
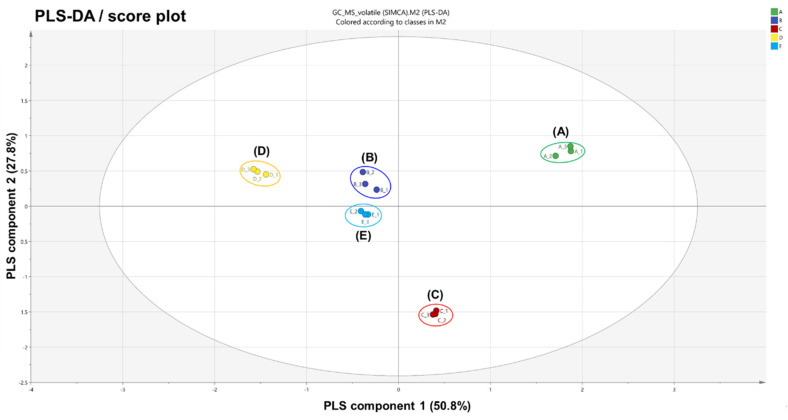
Partial least squares discriminant analysis (PLS-DA) score plot obtained from HS-SPME-GC-MS results on five variants of *Abeliophyllum distichum* flowers: (**A**) white miseon, (**B**) pink miseon, (**C**) ivory miseon, (**D**) blue miseon, (**E**) round miseon.

**Table 1 plants-10-00224-t001:** Relative contents of the detected volatile components of fresh petal tissue in five variants of *Abeliophyllum distichum* flowers using headspace solid-phase microextraction gas chromatography–mass spectrometry (HS-SPME-GC-MS) analysis.

No.	Compound	RT ^a^	LRI ^b^	Relative Content ^c^ (%) ± SD ^d^				
				White ^e^	Pink ^f^	Ivory ^g^	Blue ^h^	Round ^i^
1	isovaleraldehyde	11.50	885	1.763 ± 0.073	3.472 ± 0.240	3.641 ± 0.116	3.418 ± 0.024	2.973 ± 0.244
2	ethanol	11.91	891	6.688 ± 0.231	1.016 ± 0.039	1.133 ± 0.027	0.905 ± 0.055	2.305 ± 0.103
3	decane	15.26	1000	1.475 ± 0.103	2.023 ± 0.128	3.730 ± 0.299	1.282 ± 0.052	2.364 ± 0.187
4	5-ethyl-2,2,3-trimethylheptane	16.82	1040	0.716 ± 0.031	1.022 ± 0.129	1.453 ± 0.094	2.452 ± 0.054	1.191 ± 0.186
5	*n*-hexanal	17.67	1061	0.694 ± 0.022	0.735 ± 0.100	1.536 ± 0.011	1.104 ± 0.040	0.852 ± 0.057
6	2-butyl-3-octanol	19.22	1101	0.148 ± 0.046	0.279 ± 0.018	0.418 ± 0.054	0.218 ± 0.051	0.227 ± 0.129
7	*β*-myrcene	20.94	1146	4.232 ± 0.132	0.572 ± 0.046	0.448 ± 0.008	0.326 ± 0.005	0.610 ± 0.011
8	isoamylalcohol	21.98	1174	0.854 ± 0.013	0.951 ± 0.028	1.301 ± 0.053	0.687 ± 0.025	1.461 ± 0.139
9	limonene	22.62	1191	0.793 ± 0.031	0.176 ± 0.015	0.180 ± 0.006	0.243 ± 0.004	0.283 ± 0.007
10	2-hexenal	22.91	1198	4.831 ± 0.109	4.124 ± 0.424	8.376 ± 0.140	3.335 ± 0.098	3.407 ± 0.232
11	ocimene-1	23.54	1215	1.013 ± 0.045	0.162 ± 0.010	0.579 ± 0.007	0.072 ± 0.002	0.164 ± 0.004
12	ocimene-2	24.21	1234	3.020 ± 0.101	0.744 ± 0.058	4.792 ± 0.136	0.314 ± 0.006	0.636 ± 0.014
13	*n*-hexyl acetate	24.85	1251	1.027 ± 0.061	0.196 ± 0.019	3.825 ± 0.096	0.267 ± 0.032	0.695 ± 0.048
14	isohexanol	25.95	1281	0.210 ± 0.007	0.429 ± 0.014	0.760 ± 0.057	0.330 ± 0.007	0.403 ± 0.001
15	3-methyl-1-pentanol	26.43	1294	0.083 ± 0.005	0.101 ± 0.009	0.115 ± 0.023	0.078 ± 0.013	0.089 ± 0.013
16	2-hexenyl acetate	26.92	1307	0.133 ± 0.002	0.033 ± 0.002	0.611 ± 0.023	0.027 ± 0.001	0.108 ± 0.008
17	6-methyl-5-hepten-2-one	27.19	1315	0.102 ± 0.001	0.239 ± 0.018	0.266 ± 0.016	0.246 ± 0.002	0.287 ± 0.002
18	1-hexanol	27.32	1319	3.809 ± 0.006	2.367 ± 0.097	6.383 ± 0.250	1.794 ± 0.067	3.570 ± 0.180
19	*E*-3-eicosene	27.49	1324	0.110 ± 0.005	0.132 ± 0.011	0.159 ± 0.029	0.105 ± 0.018	0.118 ± 0.019
20	2-methyl-1-decanol	27.80	1333	0.329 ± 0.009	0.436 ± 0.019	1.341 ± 0.068	0.475 ± 0.012	0.513 ± 0.001
21	*Z*-3-hexenol	28.47	1352	1.488 ± 0.050	1.493 ± 0.057	3.262 ± 0.189	0.724 ± 0.006	1.150 ± 0.012
22	allo-ocimene	28.60	1355	0.555 ± 0.029	0.089 ± 0.007	0.334 ± 0.006	0.039 ± 0.001	0.088 ± 0.003
23	*p*-cymene	28.65	1357	0.089 ± 0.000	0.110 ± 0.010	0.198 ± 0.006	0.114 ± 0.007	0.157 ± 0.013
24	2-hexenol	29.14	1371	2.015 ± 0.019	1.551 ± 0.254	5.163 ± 0.262	0.733 ± 0.030	2.553 ± 0.275
25	nonaldehyde	29.39	1378	1.091 ± 0.031	1.498 ± 0.041	1.598 ± 0.037	1.975 ± 0.079	1.844 ± 0.036
26	acetic acid	30.42	1408	0.196 ± 0.034	0.173 ± 0.017	0.8233 ± 0.076	0.268 ± 0.019	0.214 ± 0.012
27	1-octen-3-ol	30.61	1413	0.495 ± 0.024	0.614 ± 0.009	0.959 ± 0.051	0.650 ± 0.003	0.782 ± 0.013
28	1,3-ditertarybutylbenzene	30.75	1418	3.921 ± 0.288	4.228 ± 0.323	5.504 ± 0.873	3.374 ± 0.595	5.148 ± 0.624
29	linalool oxide	31.04	1426	12.646 ± 0.180	12.625 ± 0.381	3.441 ± 0.073	2.136 ± 0.037	2.094 ± 0.060
30	benzaldehyde	33.71	1507	1.397 ± 0.043	2.843 ± 0.086	2.826 ± 0.166	2.946 ± 0.156	1.929 ± 0.099
31	l-linalool	33.88	1512	15.727 ± 0.187	3.754 ± 0.081	1.948 ± 0.079	2.758 ± 0.035	4.788 ± 0.068
32	1-octanol	34.21	1522	0.451 ± 0.012	0.495 ± 0.026	0.626 ± 0.033	0.604 ± 0.019	0.683 ± 0.034
33	lilac aldehyde A	34.32	1526	1.002 ± 0.010	1.119 ± 0.003	0.321 ± 0.005	1.588 ± 0.024	2.436 ± 0.024
34	lilac aldehyde B	34.84	1542	0.956 ± 0.007	1.097 ± 0.020	0.310 ± 0.005	1.561 ± 0.031	2.310 ± 0.041
35	lilac aldehyde C	35.06	1549	0.779 ± 0.016	0.886 ± 0.001	0.256 ± 0.007	1.289 ± 0.026	1.919 ± 0.010
36	hotrienol	35.80	1573	0.163 ± 0.018	4.198 ± 0.400	0.036 ± 0.021	2.832 ± 0.149	0.144 ± 0.013
37	lilac aldehyde D	35.88	1575	0.430 ± 0.005	0.583 ± 0.010	0.138 ± 0.002	0.761 ± 0.016	1.030 ± 0.008
38	*β*-cyclocitral	37.20	1618	0.032 ± 0.001	0.054 ± 0.001	0.109 ± 0.007	0.074 ± 0.002	0.070 ± 0.001
39	phenylacetaldehyde	37.36	1623	1.099 ± 0.072	2.441 ± 0.057	4.018 ± 0.043	2.208 ± 0.048	1.966 ± 0.067
40	2-methylbutrate	37.48	1627	0.023 ± 0.002	0.112 ± 0.010	0.294 ± 0.020	0.043 ± 0.002	0.064 ± 0.004
41	*p*-methylbenzaldehyde	37.76	1636	0.264 ± 0.038	0.598 ± 0.101	0.784 ± 0.135	0.572 ± 0.114	0.509 ± 0.153
42	*Z*-3-hexenyl angelate	38.13	1649	0.192 ± 0.004	0.043 ± 0.003	0.103 ± 0.004	0.048 ± 0.001	0.068 ± 0.005
43	salicylic aldehyde	38.58	1664	0.072 ± 0.004	0.195 ± 0.002	0.054 ± 0.001	0.461 ± 0.008	0.116 ± 0.004
44	4-oxoisophorone	39.05	1679	0.739 ± 0.022	0.414 ± 0.022	0.534 ± 0.018	1.048 ± 0.018	0.992 ± 0.053
45	lilac alcohol A	39.87	1707	0.623 ± 0.035	0.590 ± 0.050	0.189 ± 0.004	0.342 ± 0.006	0.930 ± 0.076
46	epoxylinalool	39.98	1710	1.346 ± 0.039	1.284 ± 0.083	0.329 ± 0.009	0.210 ± 0.003	0.203 ± 0.012
47	lilac alcohol B	40.51	1729	0.964 ± 0.040	1.299 ± 0.100	0.388 ± 0.008	0.525 ± 0.016	1.555 ± 0.114
48	lilac alcohol C	41.49	1763	0.852 ± 0.035	0.535 ± 0.030	0.217 ± 0.002	0.770 ± 0.019	1.376 ± 0.105
49	methyl salicylate	41.56	1766	2.125 ± 0.131	3.034 ± 0.058	2.555 ± 0.017	12.811 ± 0.298	4.683 ± 0.115
50	*β*-phenethyl acetate	42.37	1794	0.132 ± 0.001	0.281 ± 0.012	0.278 ± 0.009	0.908 ± 0.026	0.349 ± 0.006
51	lilac alcohol D	42.55	1800	1.797 ± 0.104	2.280 ± 0.205	1.044 ± 0.020	0.891 ± 0.033	3.379 ± 0.147
52	geraniol	42.74	1807	0.024 ± 0.000	0.009 ± 0.000	0.008 ± 0.001	0.009 ± 0.000	0.009 ± 0.001
53	benzenemethanol	43.71	1843	1.905 ± 0.037	1.978 ± 0.122	3.094 ± 0.052	2.124 ± 0.065	3.362 ± 0.092
54	benzeneethanol	44.79	1882	10.502 ± 0.364	25.032 ± 0.669	14.782 ± 0.212	31.453 ± 0.480	25.039 ± 0.972
55	benzyl nitrile	45.41	1905	0.008 ± 0.000	1.239 ± 0.022	0.027 ± 0.001	0.020 ± 0.001	0.013 ± 0.001
56	2-phenylbut-2-enal	45.62	1913	0.107 ± 0.003	0.265 ± 0.008	0.120 ± 0.001	0.704 ± 0.020	0.384 ± 0.008
57	*β*-lonone	46.14	1933	0.070 ± 0.004	0.109 ± 0.004	0.214 ± 0.006	0.130 ± 0.003	0.140 ± 0.005
58	2-methylphenol	46.75	1956	0.068 ± 0.003	0.083 ± 0.002	0.107 ± 0.002	0.090 ± 0.001	0.068 ± 0.002
59	*E*-nerolidol	47.92	2001	0.017 ± 0.001	0.030 ± 0.002	0.004 ± 0.000	0.016 ± 0.000	0.118 ± 0.003
60	dimethyl salicylate	48.83	2037	0.044 ± 0.004	0.031 ± 0.001	0.491 ± 0.031	0.212 ± 0.003	0.218 ± 0.008
61	ethyl linalool	50.58	2112	0.432 ± 0.007	0.461 ± 0.009	0.158 ± 0.004	0.595 ± 0.004	1.017 ± 0.016
62	eugenol	51.23	2154	0.155 ± 0.010	0.156 ± 0.020	0.623 ± 0.013	0.078 ± 0.001	0.126 ± 0.006
63	8-hydroxy-6,7-dihydrolinalool	51.62	2180	0.355 ± 0.036	0.246 ± 0.016	0.046 ± 0.007	0.893 ± 0.066	0.442 ± 0.044
64	methyl palmitate	52.69	2267	0.094 ± 0.022	0.148 ± 0.027	0.288 ± 0.032	0.192 ± 0.015	0.318 ± 0.019
65	2-hydroxylinalool	53.14	2307	0.408 ± 0.044	0.274 ± 0.023	0.160 ± 0.026	0.431 ± 0.029	0.744 ± 0.107
66	1,3-diacetylbenzene	55.68	2496	0.115 ± 0.026	0.214 ± 0.050	0.191 ± 0.045	0.112 ± 0.025	0.213 ± 0.043

^a^ Retention time (min); ^b^ linear retention indices calculated against *n*-alkanes (C_7_–C_30_); ^c^ relative contents (%); ^d^ standard error; ^e^ white miseon (*A. distichum* Nakai); ^f^ pink miseon (*A. distichum* for. *lilacinum* Nakai); ^g^ ivory miseon (*A. distichum* for. *eburneum* T.B. Lee); ^h^ blue miseon (*A. distichum* for. *viridicalycinum* T.B. Lee); ^i^ round miseon (*A. distichum* var. *rotundicarpum* T.B. Lee).

**Table 2 plants-10-00224-t002:** The variable influence on projection (VIP) values (>0.7) of the volatile components for the separation between the flowers in five variants of *Abeliophyllum distichum* in the PLS-DA-derived score plots.

No.	Compound	RT	VIP Value	No.	Compound	RT	VIP Value
1	linalool oxide	31.04	2.89	16	benzaldehyde	33.71	1.01
2	benzeneethanol	44.79	2.59	17	benzenemethanol	43.71	1.01
3	methyl salicylate	41.56	2.56	18	*n*-hexyl acetate	24.85	1.00
4	l-linalool	33.88	2.48	19	lilac aldehyde C	35.06	0.99
5	hotrienol	35.80	1.80	20	epoxylinalool	39.98	0.92
6	ethanol	11.91	1.65	21	lilac alcohol C	41.49	0.89
7	lilac alcohol D	42.55	1.60	22	1-hexanol	27.32	0.88
8	*β*-myrcene	20.94	1.34	23	2-hexenol	29.14	0.87
9	2-hexenal	22.91	1.26	24	isovaleraldehyde	11.50	0.87
10	ocimene-2	24.21	1.21	25	1,3-ditertarybutylbenzene	30.75	0.86
11	benzyl nitrile	45.41	1.18	26	isoamylalcohol	21.98	0.76
12	lilac aldehyde A	34.32	1.13	27	4-oxoisophorone	39.05	0.76
13	lilac aldehyde B	34.84	1.08	28	lilac alcohol A	39.87	0.75
14	lilac alcohol B	40.51	1.07	29	ethyl linalool	50.58	0.75
15	5-ethyl-2,2,3-trimethylheptane	16.82	1.02	30	*Z*-3-hexenol	28.47	0.75

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
