# Peer review of "Volatile Profiles of Five Variants of Abeliophyllum distichum Flowers Using Headspace Solid-Phase Microextraction Gas Chromatography–Mass Spectrometry (HS-SPME-GC-MS) Analysis"

_plants, 2021, doi:10.3390/plants10020224_

Round 1
Reviewer 1 Report
This paper reports on the characterization of the scent profiles of a family of exotic Korean flowers using HS-SPME followed by GC-MS analysis. On the positive side, this reviewer appreciates the use of a deuterated internal standard (1,2-dichlorobenzene-d4) to improve the quantitative reproducibility of the SPME extracted headspace profiles. The use of two separate multivariate analysis techniques (PCA and PLS-DA) for the characterization of the floral scent profiles is also noted as a useful way to confirm and validate the differences that exist in the relative concentrations of the set of target analytes present in these closely related plants. However, there are a number of technical aspects of the paper that are need of improvement:
- In Figure 1, it is more important to provide a single representative chromatogram of one of the Abeliophyllum distichumflowers in which the compounds listed in Table 1 are identified. It is of interest for the reader to identify the relative abundances of the compounds in the floral profile, and the chromatogram is the best way to display this information. The area in the chromatogram before the first peak of interest can also be omitted from Figure 1, since it does not contain any relevant information. The characterization of the information contained in the chromatographic profiles is more adequately addressed in Figures 2 and 3 (the multivariate analysis plots).
- The detailed mass fragmentation information in Table 2 belongs in a supplementary material Table. It is more important that Table 2 contain information on the library matching values for the compounds detected in the chromatograms. This is particularly important because no standards were reported to have been used for the confirmation of the identification of the analytes extracted from the headspace of the flowers. The library match values (in conjunction with the retention index data) enhance the level of confidence that the compounds listed in Table 2 might indeed be correctly identified.
- The odor note information provided in Table 2 is speculative at best, since it is not obtained from the experiment itself but is simply reported from flavor databases that are referenced. Unless that information was collected at the time of the experiment through the use of an olfactometer, it is not critical to the paper because these odor notes are not used in the generation of the multivariate analysis plots that differentiate the flower types.
- Reference 24 should be changed to a more accessible reference. I would suggest the paper entitled:”Linear retention indices in gas chromatographic analysis: a review” by Zellner et al. (Flavour and Fragrance Journal, 23, 297-314, 2008). Even though Kovats first introduced the concept of chromatographic retention indexing, reference 24 refers to the original work in which the indexing was obtained at isothermal conditions. The data reported in this work was obtained at temperature programmed conditions, and the adjustment to the calculation is explained in the Zellner reference.
- Finally, the quality of the writing needs to be drastically improved: there are an unusually high number of typographical errors and grammatically incorrect sentences. For example, the title of the paper itself contains a few typos: “Volatiles Profile” should be “Volatile Profiles”, “Head Space” should be “Headspace” and a number of the words in the title should be capitalized that are not (including the word “profile”). The authors should consider enlisting the services of a professional technical writer for this task due to the fact that there are so many sentences in need of adjustment. I count 23 instances in the abstract and introduction sections alone. I typically will annotate needed corrections in my reviews, but there are so many in this manuscript that it really should be properly revised before it is resubmitted.
Author Response
Dear, Editor in Chief, Plants and Reviewer 1
Thank you for your kind letter, with regard to our manuscript together with comments. We are thankful to you for the very valuable suggestions through the whole manuscript. Thank you again for your kind considerations.
We tried to revise the manuscript as much as possible in line with suggestion made by the reviewers. The corrected was expressed in blue. I am herewith enclosing improved manuscript.
Our incorporation of reviewer’s suggestions is as follow.
Reviewer 1.
This paper reports on the characterization of the scent profiles of a family of exotic Korean flowers using HS-SPME followed by GC-MS analysis. On the positive side, this reviewer appreciates the use of a deuterated internal standard (1,2-dichlorobenzene-d4) to improve the quantitative reproducibility of the SPME extracted headspace profiles. The use of two separate multivariate analysis techniques (PCA and PLS-DA) for the characterization of the floral scent profiles is also noted as a useful way to confirm and validate the differences that exist in the relative concentrations of the set of target analytes present in these closely related plants. However, there are a number of technical aspects of the paper that are need of improvement:
In Figure 1, it is more important to provide a single representative chromatogram of one of the Abeliophyllum distichum flowers in which the compounds listed in Table 1 are identified. It is of interest for the reader to identify the relative abundances of the compounds in the floral profile, and the chromatogram is the best way to display this information. The area in the chromatogram before the first peak of interest can also be omitted from Figure 1, since it does not contain any relevant information. The characterization of the information contained in the chromatographic profiles is more adequately addressed in Figures 2 and 3 (the multivariate analysis plots).
→ Thank you for your kindly indication. As reviewer 1 mentioned, Figure 1. was moved to supplementary data as Figure S1.
The detailed mass fragmentation information in Table 2 belongs in a supplementary material Table. It is more important that Table 2 contain information on the library matching values for the compounds detected in the chromatograms. This is particularly important because no standards were reported to have been used for the confirmation of the identification of the analytes extracted from the headspace of the flowers. The library match values (in conjunction with the retention index data) enhance the level of confidence that the compounds listed in Table 2 might indeed be correctly identified.
→ Thank you for your kindly indication. Actually, all the volatiles which I selected have high library match values (>85% similarity, more than 850 point in NIST library). As reviewer mentioned, quantitative MS data was moved to supplementary data (Table S1). As well library match values of each component were added in Table S1.
The odor note information provided in Table 2 is speculative at best, since it is not obtained from the experiment itself but is simply reported from flavor databases that are referenced. Unless that information was collected at the time of the experiment through the use of an olfactometer, it is not critical to the paper because these odor notes are not used in the generation of the multivariate analysis plots that differentiate the flower types.
→ Thank you for your kindly indication. As reviewer indicated, odor note information was deleted in manuscript.
Reference 24 should be changed to a more accessible reference. I would suggest the paper entitled:”Linear retention indices in gas chromatographic analysis: a review” by Zellner et al. (Flavour and Fragrance Journal, 23, 297-314, 2008). Even though Kovats first introduced the concept of chromatographic retention indexing, reference 24 refers to the original work in which the indexing was obtained at isothermal conditions. The data reported in this work was obtained at temperature programmed conditions, and the adjustment to the calculation is explained in the Zellner reference.
→ Thank you for your kindly indication. As you indicated, reference was changed reference as you recommend.
- Zellner, B.D.A.; Bicchi, C.; Dugo, P.; Rubiolo, P.; Dugo, G.; Mondello, L. Linear retention indices in gas chromatographic analysis: a review. Flavour Fragr. J. 2008, 23, 297-314.
Finally, the quality of the writing needs to be drastically improved: there are an unusually high number of typographical errors and grammatically incorrect sentences. For example, the title of the paper itself contains a few typos: “Volatiles Profile” should be “Volatile Profiles”, “Head Space” should be “Headspace” and a number of the words in the title should be capitalized that are not (including the word “profile”). The authors should consider enlisting the services of a professional technical writer for this task due to the fact that there are so many sentences in need of adjustment. I count 23 instances in the abstract and introduction sections alone. I typically will annotate needed corrections in my reviews, but there are so many in this manuscript that it really should be properly revised before it is resubmitted.
→ Actually, this manuscript was checked by a company including native speakers prior to submission. But we did our best to make the better writing including reviews’ indication.
I hope the improved version will be acceptable for publication in Plants.
Yours sincerely
Prof. Nam-In Baek

Reviewer 2 Report
In this manuscript the composition of the flower volatiles of five selected variants of Abeliophyllum distichum (Oleaceae) is presented. These experimental data were obtained by application of the HS-SPME-GC/MS. PCA analysis revealed distinct, variant-related profile of the volatiles and its correlation with morphological features of the analyzed flowers.
General comments
In this report flowers were collected for analysis at a single time point only. Authors should perform the analysis at several developmental stages of the flower to evaluate the profile of changes. It would be also nice to see the effect of plant growth conditions (insolation, humidity) on the composition of volatiles. The general significance of the observation is missing; it has to be grounded on solid data..
Quantitative MS data should be presented as a supplementary table.
Classification of the A. distichum variant should be based on the chemical data, e.g. the content of the specific pigments accumulated in the petals.
Data obtained in this manuscript are not discussed in the context of the literature, namely is the obtained profile typical for the plants of the same genus or perhaps plants in general? Do the Authors consider are volatiles unique for the A. distichum – if so could it be somehow related to a specific features of A. distichum physiology or the lifestyle of its pollinators?
Conclusions presented in this manuscript are quite limited – besides results of the additional experiments suggested below, please indicate the novelty of these data and underline their importance.
Beneficial biological effects of natural compounds isolated from A. distichum tissues have been described in several papers. Despite the fact that those metabolites were not volatile these data have to be briefly summarized in the Introdution.
Authors, please explain the biological sense of the term VIP (Table 3), spell out the abbreviation.
Are there any perspectives to use the volatile components of Abeliophyllum distichum in the industry? It would be nice to learn about such putative application in the Introduction.
Description of the biological material is far from being complete, e.g.provide the age of the flower, the age of the plant, the time of flower collection (morning – evening), plant growth conditions including insolation and humidity.
How many biological replicates were performed? How many flowers were used for each analysis?
The manuscript requires extensive revision by a native speaker. Below please find some examples:
- title: ‘Volatiles profile of five variants of Abeliophyllum distichum flowers…’ should be replaced by ‘Volatile profile of five variants of Abeliophyllum distichum flowers…’ or ‘Profile of volatiles of five variants of Abeliophyllum distichum flowers…’
- ‘Abeliophyllum distichum (Oleaceae) which comprise of not only one species but also one genus and grown only in the Korean Peninsula, has high scarcity value.’ – the meaning of this sentence is unclear while the structure is incorrect
- “Flowers fragrances and its pigments are characteristic for various insect-pollinated flowers are play a major signal to lure pollinating insects in the reproductive organs.” - ‘Flowers fragrances’ ‘ has to be replaced with ‘Flower fragrances’ moreover ‘… insect-pollinated flowers are play a major signal…’ ‘has to be replaced with ‘ …. insect-pollinated flowers; they play a role of a major signal …’
etc.
Minor remarks
Figure 1 legend – ‘’,,, (TIC) for Fresh five variants…’ – what is the meaning of ‘Fresh’ here? Were the volatiles isolated from the fresh tissue? please rephrase.
Table 1 footnote – abbreviation QI is missing – please amend
Numerous spelling and grammar mistakes have to be corrected, some examples are listed above.
Author Response
Dear, Editor in Chief, Plants and Reviewer 2
Thank you for your kind letter, with regard to our manuscript together with comments. We are thankful to you for the very valuable suggestions through the whole manuscript. Thank you again for your kind considerations.
We tried to revise the manuscript as much as possible in line with suggestion made by the reviewers. The corrected was expressed in blue. I am herewith enclosing improved manuscript.
Our incorporation of reviewer 2 suggestions is as follow.
Reviewer 2.
In this manuscript the composition of the flower volatiles of five selected variants of Abeliophyllum distichum (Oleaceae) is presented. These experimental data were obtained by application of the HS-SPME-GC/MS. PCA analysis revealed distinct, variant-related profile of the volatiles and its correlation with morphological features of the analyzed flowers.
General comments
In this report flowers were collected for analysis at a single time point only. Authors should perform the analysis at several developmental stages of the flower to evaluate the profile of changes. It would be also nice to see the effect of plant growth conditions (insolation, humidity) on the composition of volatiles. The general significance of the observation is missing; it has to be grounded on solid data.
Description of the biological material is far from being complete, e.g. provide the age of the flower, the age of the plant, the time of flower collection (morning – evening), plant growth conditions including insolation and humidity.
→ Thank you for your valuable indication. As you mentioned, we added information of biological materials in manuscript.
“Approximately 30 g of raw floral tissue of five variants of A. distichum was collected between 10:00 a.m. to 10:30 a.m., on 25th March 2017 (12 ± 1°C; 46 ± 2% relative humidity). The moistened samples were immediately transported to the laboratory.”
Quantitative MS data should be presented as a supplementary table.
→ As reviewer 2 indicated, quantitative MS data was moved to supplementary data (Table S1.)
Classification of the A. distichum variant should be based on the chemical data, e.g. the content of the specific pigments accumulated in the petals.
→ Thank you for your comment. I really agree to your indication. But the purpose of this study was identified and classification of five variants of A. distichum via their volatile components. As you mentioned, the color of petals was related with its pigments such as carotenoids and flavonoids. Through follow-up research, we will study detailed classification using combined metabolomics tool, NMR, LC/MS, GC/MS. Please we hope that this has been appropriate and acceptable answer to your comment.
Data obtained in this manuscript are not discussed in the context of the literature, namely is the obtained profile typical for the plants of the same genus or perhaps plants in general? Do the Authors consider are volatiles unique for the A. distichum – if so could it be somehow related to a specific features of A. distichum physiology or the lifestyle of its pollinators?
→ Thank you for your kindly indication. In my opinion, scents are enough to be fingerprint of the fragrant plant. As shown in our results, some unique volatiles, lilac aldehydes and lilac alcohols, are detected in this plant as major components. These volatiles are specifically observed in some of Oleaceae plants. As you know, volatiles, like other metabolites, vary in content according to external condition such as temperature and humidity. Thus, we have taken and compared samples under the same conditions at the same location to exclude these factors. Therefore, we think that these data have significant meanings. Usually, flowers attract insects or animals with their pigments or scents for reproduction. Although the volatile components may not directly affect their physiology, we thought variants which have the relatively small variety of pigments would have different scents from other varieties to survive. Please we hope that this has been appropriate and acceptable answer to your comment.
Conclusions presented in this manuscript are quite limited – besides results of the additional experiments suggested below, please indicate the novelty of these data and underline their importance.
→ Thank you for your kindly indication. As you indicated, the novelty and importance of this study were added in conclusion section. Please we hope that this has been appropriate and acceptable answer to your comment.
At present, five variants of this plant have been reported: white miseon (A. distichum Nakai), pink miseon (A. distichum for. lilacinum Nakai), ivory miseon (A. distichum for. eburneum T. B. Lee), blue miseon (A. distichum for. viridicalycinum T. B. Lee), and round miseon (A. distichum var. rotundicarpum T. B. Lee) [26-28]. The variants were classified based only on morphological characteristics like the color of the petals and sepals or the shapes of the fruit. There are many opinions on the taxonomic identities of variants of A. distichum, even some documents suggest that each variant was same taxa [33]. Accordingly, phytochemical investigation and chemical mapping for the variants should be valuable.
In this study, we performed volatile components analysis of A. distichum flowers and multivariate data analyses to reveal the relation between fragments emitted from five variants of A. distichum flowers with their morphological characteristics. 66 volatile components of this plant identified and quantified by HS-SPME-GC/MS. As a results of multivariate data analyses, almost volatiles showed unique patterns according to morphological characteristics, especially color of petals. These results suggest that morphological characteristics of each variant were related to the content of volatiles. Also further studies are needed to understand chemical origin of the morphological characteristics of each variant, to reveal the dissimilarities of primary and secondary metabolites of this plants.
- Kim, D.K.; Park, K.R.; Kim, J.H. A taxonomic study of Abeliophyllum Nakai (Oleaceae) based on RAPD analysis. Korean J. Plant Res. 2002, 15, 26-35.
Beneficial biological effects of natural compounds isolated from A. distichum tissues have been described in several papers. Despite the fact that those metabolites were not volatile these data have to be briefly summarized in the Introdution.
→ Thank you for your valuable indication. As you mentioned, we described additional information about biological effects of natural compounds isolated from A. distichum as below.
“Due to its high scarcity and ecological and geographic value, the study of A. distichum has rarely been carried out compared with other plants. Previous phytochemical studies of these plants have focused on just the leaves, and they reported that four phenylethanoid glycosides and two flavonoids are components of this plants [6-8]. These compounds have been reported to exert anti-oxidation, anti-inflammation, anti-hypertension, anti-diabetes, and neuroprotection [7-10].”
- Kuwajima, H.; Takahashi, M.; Ito, M.; Wu, H.X.; Takaishi, K.; Inoue, K. A quinol glucoside from Abeliophyllum distichum. Phytochem. 1993, 33, 137-139.
- Oh, H.; Kang, D.G.; Kwon, T.O.; Jang, K.K.; Chai, K.Y.; Yun, Y.G.; Chung,T.; Lee, H.S. Four glycosides from the leaves of Abeliophyllum distichum with inhibitory effects on angiotensin converting enzyme. Phytother. Res. 2003, 17, 811-813.
- Li, H.M.; Kim, J.K.; Jang, J.M.; Cui,B.; Lim, S.S. Analysis of the inhibitory activity of Abeliophyllum distichum leaf constituents against aldose reductase by using high-speed counter current chromatography. Arch. Pharm. Res. 2013, 36, 1104-1112.
- Lee, Y.G.; Lee, H.; Jung, J.W.; Seo, K.H.; Lee, D.Y.; Kim, H.G.; Ko, J.H.; Lee, D.S.; Baek, N.I. Flavonoids from Chionanthus retusus (Oleaceae) flowers and their protective effects against glutamate-induced cell toxicity in HT22 cells. J. Mol. Sci. 2019, 20, 3517.
- Lee, Y.G.; Seo, K.H.; Lee, D.S.; Gwag, J.E.; Kim, H.G.; Ko, J.H.; Park S.H.; Lee, D.Y.; Baek, N.I. Phenylethanoid glycoside from Forsythia koreana (Oleaceae) flowers shows a neuroprotective effect. Braz. J. Bot. 2018, 41, 523-528.
Authors, please explain the biological sense of the term VIP (Table 3), spell out the abbreviation.
→ As indicated, full name of VIP was described in manuscript.
Based on variable influence on projection (VIP, Table 2), 30 volatiles from A. distichum flowers are higher than 0.7, and 18 scents showed VIPs larger than 1.0, which is the most influential value for this model [23].
Are there any perspectives to use the volatile components of Abeliophyllum distichum in the industry? It would be nice to learn about such putative application in the Introduction.
→ Thank you for your indication. As reviewer mentioned, we described additional information about industrial perspectives to use the A. distichum flowers and its volatiles in introduction. Please we hope that this has been appropriate and acceptable answer to your comment.
“From this reason, Korean government designated A. distichum as endangered species until a recent date, even some of its habitats as natural monument to preserve their natural habitats [6]. Due to its high scarcity and ecological and geographic value, the study of A. distichum has rarely been carried out compared with other plants. But in 2017, A. distichum was deleted in list of endangered species by Ministry of Environment [6], due to develop of mass breeding technique of its plants [7,8]. Previous phytochemical studies of these plants have focused on just the leaves, and they reported that four phenylethanoid glycosides and two flavonoids are components of this plants [9-11]. These compounds have been reported to exert anti-oxidation, anti-inflammation, anti-hypertension, anti-diabetes, and neuroprotection [10-13].
Also, this plant is known to have attractively and strongly fragrances [14,15]. Even though, A. distichum has good fragrance, rarely study has been reported for the analysis of the volatiles. According to this aspect, its volatiles can be manufactured to fragrance oil for the various industry such as aromatherapy, perfumes, cosmetics. So our further study will be focused on to reveal the biological effect and standardize its content for cosmeceutical industry. Accordingly, volatiles analysis of A. distichum has high value both of researchial and industrial perspectives.”
How many biological replicates were performed? How many flowers were used for each analysis?
→ Thank you for your indication. I missed important information of this experiments. And as reviewer indicated, we added missed information in manuscript.
“Approximately ten flowers were used for each sample.”
“Analysis of each sample was performed three times, respectively.”
The manuscript requires extensive revision by a native speaker. Below please find some examples:
- title: ‘Volatiles profile of five variants of Abeliophyllum distichum flowers…’ should be replaced by ‘Volatile profile of five variants of Abeliophyllum distichum flowers…’ or ‘Profile of volatiles of five variants of Abeliophyllum distichum flowers…’
- ‘Abeliophyllum distichum (Oleaceae) which comprise of not only one species but also one genus and grown only in the Korean Peninsula, has high scarcity value.’ – the meaning of this sentence is unclear while the structure is incorrect
- “Flowers fragrances and its pigments are characteristic for various insect-pollinated flowers are play a major signal to lure pollinating insects in the reproductive organs.” - ‘Flowers fragrances’ ‘has to be replaced with ‘Flower fragrances’ moreover ‘… insect-pollinated flowers are play a major signal…’ ‘has to be replaced with ‘ …. insect-pollinated flowers; they play a role of a major signal …’ etc.
Numerous spelling and grammar mistakes have to be corrected, some examples are listed above.
→ Actually, this manuscript was checked by a company including native speakers prior to submission. But we did our best to make the better writing including reviews’ indication. Please we hope that this has been appropriate and acceptable answer to your comment.
Minor remarks
Figure 1 legend – ‘’,,, (TIC) for Fresh five variants…’ – what is the meaning of ‘Fresh’ here? Were the volatiles isolated from the fresh tissue? please rephrase.
→ Thank you for your important comment. The meaning of 'fresh’ was describe in M&M section. And as reviewer indicated, “fresh flower of five variants of Abeliophyllum distichum flowers” was changed to “raw floral tissue of five variants of Abeliophyllum distichum”
Table 1 footnote – abbreviation QI is missing – please amend
→ Thank you for your kind indication. As you mentioned, we added abbreviation of QI like below.
‘dquantification ion (m/z), Specific mass ion used for quantification’
I hope the improved version will be acceptable for publication in Plants.
Yours sincerely
Prof. Nam-In Baek

Reviewer 3 Report
The study performed volatile components analysis of A. distichum flowers and multivariate data analyses to reveal the relation between fragments emitted from five variants of A. distichum flowers with their morphological characteristics.
Results obtained are complex. The multivariate analysis permitted to have an integrated image of the importance of these findings.
Here are my observations:
- For the industrial application of these findings, the amount of each compound is extremely important to be specified. So please complete Table 2 with these data. Address in the Results and discussion section the related information. Otherwise, the paper is incomplete.
- The conclusion section must not repeat the Abstract and must contain the relevant information presented in the manuscript. Point out the novelty brought by these findings by focusing on results obtained, further studies authors will want to address (optional) and the relevance of this study for the industrial sector.
Author Response
Dear, Editor in Chief, Plants and Reviewer 3
Thank you for your kind letter, with regard to our manuscript together with comments. We are thankful to you for the very valuable suggestions through the whole manuscript. Thank you again for your kind considerations.
We tried to revise the manuscript as much as possible in line with suggestion made by the reviewers. The corrected was expressed in blue. I am herewith enclosing improved manuscript.
Our incorporation of reviewer’s suggestions is as follow.
Reviewer 3.
The study performed volatile components analysis of A. distichum flowers and multivariate data analyses to reveal the relation between fragments emitted from five variants of A. distichum flowers with their morphological characteristics.
Results obtained are complex. The multivariate analysis permitted to have an integrated image of the importance of these findings.
Here are my observations:
For the industrial application of these findings, the amount of each compound is extremely important to be specified. So please complete Table 2 with these data. Address in the Results and discussion section the related information. Otherwise, the paper is incomplete.
→ Thank you for your kindly indication. As you indicated, relative content values of the detected volatile components of each variant were added in Table 1.
The conclusion section must not repeat the Abstract and must contain the relevant information presented in the manuscript. Point out the novelty brought by these findings by focusing on results obtained, further studies authors will want to address (optional) and the relevance of this study for the industrial sector.
→ Thank you for your kindly indication. As you indicated, the novelty and importance of this study were added in conclusion section. Please we hope that this has been appropriate and acceptable answer to your comment.
At present, five variants of this plant have been reported: white miseon (A. distichum Nakai), pink miseon (A. distichum for. lilacinum Nakai), ivory miseon (A. distichum for. eburneum T. B. Lee), blue miseon (A. distichum for. viridicalycinum T. B. Lee), and round miseon (A. distichum var. rotundicarpum T. B. Lee) [26-28]. The variants were classified based only on morphological characteristics like the color of the petals and sepals or the shapes of the fruit. There are many opinions on the taxonomic identities of variants of A. distichum, even some documents suggest that each variant was same taxa [33]. Accordingly, phytochemical investigation and chemical mapping for the variants should be valuable.
In this study, we performed volatile components analysis of A. distichum flowers and multivariate data analyses to reveal the relation between fragments emitted from five variants of A. distichum flowers with their morphological characteristics. 66 volatile components of this plant identified and quantified by HS-SPME-GC/MS. As a results of multivariate data analyses, almost volatiles showed unique patterns according to morphological characteristics, especially color of petals. These results suggest that morphological characteristics of each variant were related to the content of volatiles. Also further studies are needed to understand chemical origin of the morphological characteristics of each variant, to reveal the dissimilarities of primary and secondary metabolites of this plants.
- Kim, D.K.; Park, K.R.; Kim, J.H. A taxonomic study of Abeliophyllum Nakai (Oleaceae) based on RAPD analysis. Korean J. Plant Res. 2002, 15, 26-35.
I hope the improved version will be acceptable for publication in Plants.
Yours sincerely
Prof. Nam-In Baek

Round 2
Reviewer 2 Report
Authors’ effort to improve the manuscript is appreciated. Additional information together with appropriate references are included in the revised manuscript.
Unfortunately no new experimental data are presented (please refer to my first comment of the previous review). The manuscript postulates variant-specific profile of the volatiles emitted by A. distichum flowers. In the opinion of this reviewer growth conditions and the developmental stage of the flower surely affect the composition of emitted volatiles. It should be verified experimentally whether observed differences are modulated by these factors. Lack of these data questions to scientific significance of this manuscript.
Some English errors have been corrected but still a big room for improvement remains here - the manuscript requires revision by a native speaker.
Author Response
Dear, Editor in Chief, Plants and Reviewer 2
Thank you for your kind letter, with regard to our manuscript together with comments. We are thankful to you for the very valuable suggestions through the whole manuscript. Thank you again for your kind considerations.
We tried to revise the manuscript as much as possible in line with suggestion made by the reviewers. The corrected was expressed in blue. I am herewith enclosing improved manuscript.
Our incorporation of reviewer 2 suggestions is as follow.
Reviewer 2.
Authors’ effort to improve the manuscript is appreciated. Additional information together with appropriate references are included in the revised manuscript.
Unfortunately no new experimental data are presented (please refer to my first comment of the previous review). The manuscript postulates variant-specific profile of the volatiles emitted by A. distichum flowers. In the opinion of this reviewer growth conditions and the developmental stage of the flower surely affect the composition of emitted volatiles. It should be verified experimentally whether observed differences are modulated by these factors. Lack of these data questions to scientific significance of this manuscript.
→ Thank you for your comment. I really agree to your indication which the variation of volatiles is affected and associated with developmental stage and its growth conditions. But according to the purpose of this study to classify clearly of five variants of A. distichum via their volatile components, all variations (plant age, developmental stage, humidity, temperature, collecting region) were not considered in this experiment. To minimize variables in this experiment, our assay was conducted for all variations (plant age, developmental stage, humidity, temperature, collecting region) fixed to one condition. Therefore, samples which reviewer mentioned are not prepared for each season and developmental stage. Also it is currently impossible to collect them because it is not the time when flowers bloom. Through follow-up research, we will study variation of volatiles according to their growth conditions and developmental stage. Please we hope that this has been appropriate and acceptable answer to your comment.
Some English errors have been corrected but still a big room for improvement remains here - the manuscript requires revision by a native speaker.
→ Thank you for your indication. As you mentioned, this manuscript was revised by a native speaker.
I hope the improved version will be acceptable for publication in Plants.
Yours sincerely
Prof. Nam-In Baek

Round 3
Reviewer 2 Report
I do not have any new comment